# Signalling Pathways Mediating the Effects of CD40-Activated CD40L Reverse Signalling on Inhibitory Medium Spiny Neuron Neurite Growth

**DOI:** 10.3390/cells10040829

**Published:** 2021-04-07

**Authors:** Paulina Carriba, Alun M. Davies

**Affiliations:** School of Biosciences, Cardiff University, Museum Avenue, Cardiff CF10 3AX, UK; daviesalun@cardiff.ac.uk

**Keywords:** CD40L-reverse signalling, neurite growth, inhibitory neuron, PKC, JNK, ERK1/2, Syk

## Abstract

CD40-activated CD40L-mediated reverse signalling is a major physiological regulator of neurite growth from excitatory and inhibitory neurons in the developing central nervous system (CNS). Whereas in excitatory pyramidal neurons, CD40L reverse signalling promotes the growth and elaboration of dendrites and axons, in inhibitory GABAergic striatal medium spiny neurons (MSNs), it restricts neurite growth and branching. In pyramidal neurons, we previously reported that CD40L reverse signalling activates an interconnected and interdependent signalling network involving protein kinase C (PKC), extracellular regulated kinases 1 and 2 (ERK1/2), and c-Jun *N*-terminal kinase (JNK) signalling pathways that regulates dendrite and axon growth. Here, we have studied whether these signalling pathways also influence neurite growth from striatal inhibitory MSNs. To unequivocally activate CD40L reverse signalling, we treated MSN cultures from CD40-deficient mice with CD40-Fc. Here, we report that activation of CD40L reverse signalling in these cultures also increased the phosphorylation of PKC, ERK1/2, and JNK. Using pharmacological activators and inhibitors of these signalling pathways singularly and in combination, we have shown that, as in pyramidal neurons, these signalling pathways work in an interconnected and interdependent network to regulate the neurite growth, but their functions, relationships, and interdependencies are different from those observed in pyramidal neurons. Furthermore, immunoprecipitation studies showed that stimulation of CD40L reverse signalling recruits the catalytic fragment of Syk tyrosine kinase, but in contrast to pyramidal neurons, PKC does not participate in this recruitment. Our findings show that distinctive networks of three signalling pathways mediate the opposite effects of CD40L reverse signalling on neurite growth in excitatory and inhibitory neurons.

## 1. Introduction

CD40-activated CD40L reverse signalling influences the growth and elaboration of neural processes in several kinds of neurons in both the peripheral and the central nervous systems. In the central nervous system (CNS), CD40-activated CD40L reverse signalling has strikingly opposite effects on the size and elaboration of the neurites (axon and dendrites) of excitatory and inhibitory neurons. Elimination of CD40/CD40L signalling in *Cd40*^−/−^ mice results in marked and opposite phenotypic changes in these neurons both in vivo and in vitro. Compared with wild-type mice, the neurites of the excitatory glutamatergic pyramidal neurons of the hippocampus are markedly stunted, whereas those of the inhibitory GABAergic medium spiny neurons (MSNs) of the striatum are much larger and more exuberant. In vitro experiments demonstrated that these phenotypic changes in both kinds of neurons are due to the elimination of CD40-activated CD40L-mediated reverse signalling rather than to the elimination of CD40L-activated CD40 forward signalling. Indeed, CD40 forward signalling does not play any role in the regulation of the growth and elaboration of neural processes in MSNs and pyramidal neurons [1]. These morphological effects mediated by CD40L-reverse signalling are dependent on protein kinase C (PKC); with PKCβ involved in pyramidal neurons and PKCγ in MSNs [1].

In pyramidal neurons, we recently reported that the activation of CD40L reverse signalling activates PKC, ERK1/2 (extracellular regulated kinases 1 and 2), and JNK (c-Jun *N*-terminal kinase) and that these proteins function as an interdependent and interconnected signalling network regulating the growth of dendrites and axons [2]. Previous work has shown that these signalling pathways are individually involved in regulating the growth of neural processes mediated by several factors from a variety of neurons, mediating either enhancement or suppression of axon and/or dendrite growth [3,4,5,6,7,8,9,10,11,12]. Our findings in pyramidal neurons suggested instead that these signalling pathways do not function in a simple linear sequence, but rather they act interconnected and in a distinctively regulated network. Because in MSNs, CD40L reverse signalling also regulates the growth of neurites, the aim of the present study was to investigate the downstream mechanism in these inhibitory neurons following the activation of CD40L-mediated reverse signalling. We first analyzed whether PKC, JNK, and ERK1/ERK2 are also involved as in excitatory neurons. By analysing the function of these signalling proteins and their relationships, we determined the downstream signalling in these inhibitory neurons, identifying the similarities and differences between these two kinds of neurons that account for the distinctive neurite growth responses of excitatory and inhibitory neurons to CD40-activated CD40L-mediated reverse signalling.

For these studies, we used MSN neurons cultured from CD40-deficient mice to eliminate any endogenous CD40/CD40L signalling, and activated CD40L reverse signalling by treating the neurons with CD40-Fc. CD40-Fc protein is a chimeric protein consisting of the extracellular domain of CD40 linked to the Fc part of the human IgG1 that is able to activate CD40L-reverse signalling [1,2,13,14]. Wild-type neurons are not useful for studying intracellular signalling activated by CD40L reverse signalling because treating wild-type neurons with CD40-Fc does not affect neurite length, as previously observed in several populations of neurons where the control of the dendrite or axon growth is influenced by CD40L reverse signalling [1,13]. Moreover, treating wild-type neurons with soluble CD40 (sCD40L) could not only activate CD40 forward signalling, but could also compete with the endogenous CD40L blocking the CD40L reverse signalling [1,13], making the results difficult to interpret. The activation of CD40L reverse signalling in cultured MSNs increased the phosphorylation levels of PKC, ERK1/2, and JNK. By using pharmacological activators and inhibitors of these signalling pathways singularly and in combination, we showed that these signalling pathways mediate the inhibitory effect of CD40L reverse signalling on neurite growth by functioning as a distinctive interconnected and interdependent network. Whereas the morphological assays showed the hierarchy by which these signalling pathways act, the phosphorylation studies provided their differential and distinctive regulation. As in hippocampal pyramidal neurons, the activation of CD40L reverse signalling also involves the recruitment of Syk, but unlike in these neurons, the receptor complex formed by CD40L and Syk in MSNs does not comprise PKC. These findings show that distinctive networks of three signalling pathways mediate the opposite effects of CD40L reverse signalling on neurite growth in excitatory and inhibitory neurons.

## 2. Materials and Methods

### 2.1. Mice

Mice breeding was approved by the Cardiff University Ethical Review Board and was performed within the guidelines of the Home Office Animals (Scientific Procedures) Act, 1986. Mice were housed in a 12 h light-dark cycle with access to food and water ad libitum. *Cd40* null mutant mice in a C57BL6/J background were purchased from The Jackson Laboratory (Bar Harbour, ME, USA). These mice were back-crossed into a CD1 background. *Cd40*^+/−^ mice were crossed to generate *Cd40*^−/−^ mice from which cultures were established.

### 2.2. Neuron Culture

To prepare the primary medium spiny neuron (MSN) cultures, striatal primordia were dissected from embryonic day 14 (E14) mouse fetuses and were triturated to produce a single cell suspension following trypsin digestion (Worthington, Lakewood, CA, USA) and DNase I treatment (Roche Applied Science, East Sussex, UK). Neurons were plated at a density of 15,000 cells/cm^2^ for the morphological analysis and at 20,000 cells/cm^2^ for western blot experiments in plastic dishes coated with poly-L-lysine (Sigma-Aldrich, Dorset, UK). Neurons were cultured with Neurobasal A (Invitrogen, Paisley, UK) supplemented with 2% NeuroCult SM1 neuronal supplement (StemCell, Cambridge, UK), 1% Foetal Calf Serum (FCS) (Sigma-Aldrich, Dorset, UK), 100 units/mL penicillin, and 100 μg/mL streptomycin (Gibco BRL, Crewe, UK). To avoid large astrocyte proliferation, at 3 DIV, half of the medium was replaced with the same medium as before but without FCS, and from 7 DIV, the medium was replaced every 3–4 days with medium without FCS. The cultures were incubated at 37 °C in a humidified atmosphere containing 5% CO_2_.

The cultures were treated with the following reagents as indicated in the text: CD40-Fc (1 μg/mL, ALX-522-016-C050), Fc protein (1 μg/mL, ALX-203-004-C050), SP600125 (1 μM, BML-EI305-0010) and U0126 (1 μM, BML-EI282-0001) from Enzo Life Sciences; Anisomycin (50 nM, cat. no. 1290), Fisetin (1 μM, cat. no. 5016), Go 6983 (500 nM, cat. no. 2285), and U0124 (1 μM, cat. no. 1868) from Tocris Biosciences; phorbol-12-myristate-13-acetate (PMA) (500 nM, MERCK, cat. no. 524400). Fc and CD40-Fc were reconstituted with sterile H_2_O; the rest were reconstituted in dimethyl sulfoxide (DMSO) and subsequently diluted in culture medium to the concentrations indicated. No differences were observed between control cultures that received an equivalent level of DMSO, untreated cultures, and cultures treated with Fc. None of the treatments altered the neuronal viability or cell density. During and after treatments, the neurons showed healthy morphologies without any sign of cell death or damage.

### 2.3. Analysis of Neurite Morphology

For the analysis of neurite morphology, treatments were performed the next day after seeding the neurons. Treatments were added to the fresh medium when the medium was partially replaced. MSN cultures were fixed with 4% paraformaldehyde after 10 days in vitro, after which they were permeabilizated with 0.1% Triton-X100 and blocked with 1% bovine serum albumin (BSA) before labelled with anti-DARPP-32 (1:400 Cell Signaling Technology, Danvers, MA, USA) to identify MSNs. Analysis of neuronal morphology was studied after double labelling with anti-βIII tubulin (1:1500; chicken ab41489, AbCam, Cambridge, UK). Secondary antibodies used to visualize labelling neurons were fluorescent Alexa antibodies (1:500; Thermofisher, Cambridge, UK, A-21206 and A-11042). Labelled neurons were visualized using a Zeiss LSM710 confocal microscope.

Neurite length was assessed using Fiji (ImageJ) software with the semiautomated plugin Simple Neurite Tracer [15]. The mean and standard errors of the measurements from at least three independent experiments were plotted.

### 2.4. Prediction of Protein-Protein Interactions with STRING

As an in silico approach to determine possible protein-protein interactions (PPI) between CD40L and PKCγ, we used STRING database (STRING: functional protein-protein interaction networks; https://string-db.org/ (accessed on 12 March 2021)). All possible interactions including text-mining, experiments, databases, co-expression, neighborhood, gene fusion, and co-occurrence with reliability scores more than 0.4 (0.4 medium confidence) were analyzed for mouse CD40L and PKCγ. Appendix A shows all possible PPI for CD40L, PKCγ, and all that they share in common.

### 2.5. CD40L Pull Down

Protein interactions of CD40L after binding CD40-Fc were analyzed in 9 days in vitro MSNs cultures from *Cd40*^−/−^ mice treated for 30 min with either Fc or CD40-Fc (1 μg/mL). The neuron cells were then washed with ice-cold PBS, harvested, and lysed in ice-cold triton lysis buffer (NaCl 150 mM, EDTA 10 mM, Tris-HCl 10 mM pH 7.4, 1% Triton X-100, and protease and phosphatase inhibitor cocktail mix (Protease/Phosphatase inhibitor cocktail, 5872, Cell Signaling, London, UK)). After lysate clearance by centrifugation and quantify for equal concentration, Fc fragments were pulled down from the supernatant by incubation overnight on an orbital shaker at 4 °C with protein G-Sepharose beads (Protein G Sepharose Fast Flow, P3296, Sigma, Dorset, UK) previously blocked with 5% BSA. The beads were washed 5× with ice-cold triton lysis buffer. Complexes were collected with elution citrate 0.1 M pH 2.5 buffer. The pH was adjusted by adding 1/6 neutralizing Tris HCl 1 M pH 8.5 buffer, and after adding Laemmli buffer, the samples were boiled for analysis by immunoblotting.

### 2.6. Immunoblotting

For the immunoblotting, dissected striatal primordia were placed in triton lysis buffer supplemented with protease and phosphatase inhibitor cocktail mix (Protease/Phosphatase inhibitor cocktail, 5872, Cell Signaling, London, UK). The tissue was disaggregated using a pellet pestle until completely homogenizated. For the cultured neurons from *Cd40*^−/−^ mice after the indicated treatments, the neurons were scraped from the plates in ice-cold PBS, collected by centrifugation, and resuspended in ice-cold triton lysis buffer. For preparation of cytosolic and nuclear extracts, the neurons were resuspended in buffer A containing 10 mM Hepes pH 7.9, 1.5 mM MgCl_2_, 10 mM KCl, 0.5 mM dithiothreitol (DTT) with protease and phosphatase inhibitor cocktail mix. Nonidet P40 was added to a final concentration of 0.6% and vortexed for 10 s. Nuclei were separated from the cytosolic extracts by centrifugation. The nuclei were then washed once with buffer A and were incubated with buffer B that contained 20 mM Hepes pH 7.9, 25% glycerol, 400 nM NaCl, 1 mM ethylenediamine tetraacetic acid (EDTA), 0.5 mM dithiothreitol (DTT), and protease and phosphatase inhibitor cocktail mix for 30 min with gentle rocking at 4 °C. The suspensions were centrifugated at 15,000× *g* for 15 min at 4 °C. Equal quantities of protein were separated on 10% SDS-PAGE gels and were transferred to polyvinylidene difluoride (PVDF) membranes (Immobilon-P, Millipore, Dorset, UK). The blots were probed with anti-phosphoPKC^Thr514^ (1:1000; rabbit 9379), anti-phospho-p44/p42 MAPK (ERK1/2)^Thr202/Tyr204^ (1:1000; rabbit 9101), anti-p44/p42 (ERK1/2) (1:1000; mouse 9107), anti-phospho-SAPK/JNK^Thr183/Tyr185^ (1:1000; rabbit 4671), anti-Syk (1:1000; rabbit 2712) that detects the 72 kDa and the 40 kDa because it is generated using an epitope at the carboxyl terminal, and anti-PKCγ (1:1000; rabbit 43806) from Cell Signaling, London, UK; anti-PKC (1:1000; mouse clone M110 05-983, MERCK, Dorset, UK), anti-CD40L (1:700; rabbit ab2391, AbCam, Cambridge, UK), anti-PKCβ (1:1000; rabbit SAB4502358, Sigma, Dorset, UK), and anti-βIII tubulin (1:90,000; mouse MAB1195, R&D, Abingdon, UK). Binding of the primary antibodies was visualized with HRP-conjugated donkey anti-rabbit or anti-mouse secondary antibodies (1:5000; rabbit W4011, mouse W4021, Promega, Southampton, UK) and EZ-ECL kit Enhanced Chemiluminescence Detection Kit (Biological Industries, Geneflow Limited, Staffordshire, UK).

## 3. Results

### 3.1. Activation of CD40L Reverse Signalling Phosphorylates PKC, ERK, and JNK

Because CD40-activated CD40L-mediated reverse signalling enhances phosphorylation of PKC, ERK1/2, and JNK in hippocampal pyramidal neurons, we initially determined whether these three signalling pathways were also activated by CD40L reverse signalling in MSNs. We used western blotting to assess the phosphorylation levels of these proteins in MSN cultures after activating CD40L reverse signalling with CD40-Fc. To prevent CD40L reverse signalling in the absence of CD40-Fc, MSN cultures were established from *Cd40*^−/−^ mice. These cultures were treated with either CD40-Fc (Figure 1a) or Fc as a control (Figure 1b) for the times indicated collecting the lysates after a total of 9 days in vitro. Lysates were analyzed by western blotting for the levels of phospho-PKC^Thr514^, phospho-ERK1/ERK2^Thr202/Tyr204^, and phospho-JNK^Thr183/Tyr185^. Anti-βIII tubulin was used for normalizing the relative phosphorylation of pPKC and pERK1/pERK2 (cytosolic fraction), and naphthol blue for pJNK (nuclear fraction).

There were significant increases in the levels of all three phospho-proteins in neurons treated with CD40-Fc compared with unchanged basal levels detected in neurons treated with the control Fc (Figure 1). There was a pronounced peak of all three phospho-proteins 20 min after treatment with CD40-Fc, but after this point, there were some differences in the time course of phosphorylation (Figure 1c). After a peak at 20 min, the level of phospho-PKC decreased to a steady level similar to that observed in Fc-treated cultures, whereas the decreases observed in phospho-ERK1/phospho-ERK2 and phospho-JNK were less pronounced with a second less pronounced peak at 8 h. After this second peak at 8 h, the phosphorylation levels of phospho-ERK1/phospho-ERK2 induced by CD40-Fc remained elevated in relationship to those in neurons treated with Fc. The phosphorylation level of phospho-JNK gradually decreased after this time, reaching baseline by 30 h. These results indicate that CD40L-mediated reverse signalling increases the phosphorylation and activation of all three signalling pathways with a distinctive kinetic profile for each phospho-protein.

### 3.2. Pharmacological Manipulation of Neurite Growth from Cd40^−/−^ Cultured Striatal GABAergic Neurons

We used specific activators and inhibitors of the PKC, JNK, and ERK1/ERK2 signalling pathways to investigate the function of these signalling proteins in the control of neurite growth in response to CD40L reverse signalling. The pharmacological reagents plus either CD40-Fc or Fc were added to the neurons 24 h after plating. After a further 9 days in vitro, the neurons were fixed and double immunolabelled for the analysis neurite length with anti-βIII tubulin (red) and an antibody to dopamine and cyclic AMP-regulated protein (DARPP-32) (green) to positively identify MSNs [16]. As we previously reported, MSNs cultured from E14 *Cd40*^−/−^ embryos replicate the in vivo phenotype with exuberant and larger neurites, and the activation of CD40L reverse signalling by treatment with a CD40-Fc chimera restores the wild-type phenotype [1] (Figure 2a). Representative images and quantification of the total neurite length of Fc-treated (white bars) and CD40-Fc-treated (grey bars) *Cd40*^−/−^ MSNs simultaneously treated with activators or inhibitors of PKC (Figure 2b,c), JNK (Figure 2d,e), and ERK1/ERK2 (Figure 2f,g) are shown.

In cultures of striatal MSNs from *Cd40*^−/−^ embryos, the exuberant neurites observed in Fc-treated cultures were significantly suppressed when phorbol-12-myristate 13-acetate (PMA) was added (Figure 2b,c). PMA is a PKC activator analogue of diacylglycerol that activates conventional PKCs and novel PKCs [17]. This growth suppression mimicked that observed when the neurons were treated with CD40-Fc to activate the reverse signalling. When CD40L reverse signalling was additionally activated with CD40-Fc, the simultaneous addition of PMA had no effect on the extent of neurite growth (Figure 2b,c). The pan-PKC inhibitor Go6983 prevented the in vitro rescue of the neurite phenotype of CD40-deficient neurons when CD40L reverse signalling was activated with CD40-Fc (grey bar), but had no effect on neurite length in control neurons treated with Fc (white bar) (Figure 2b,c). There were no significant differences in total neurite length between MSNs treated with CD40-Fc plus Go6983 and neurons treated with either Fc protein or Fc plus Go6983 (Figure 2c). These results reaffirm the importance of PKC in the effect of CD40-activated CD40L mediates on neurite growth suppression from developing MSNs.

The activation of JNK was prevented by treating the neurons with the anthrapyrazolone SP600125 [18]. SP600125 prevented rescue of the enhanced neurite growth phenotype of *Cd40*^−/−^ by CD40-Fc (Figure 2d,e, grey bar), with no significant difference in the total neurite length of *Cd40*^−/−^ MSNs treated with Fc and MSNs treated with Fc plus SP600125 (Figure 2d,e white bar). However, a small statistically significant reduction in neurite length was observed with SP600125 when CD40L reverse signalling was activated with CD40-Fc, indicating that when CD40L reverse signalling is activated, the inhibition of JNK partly prevented neurite growth (Figure 2e). JNK was activated by anisomycin (Ani), a translational inhibitor secreted by Streptomyces [19]. Ani did not interfere with rescue of the enhanced neurite growth phenotype of *Cd40*^−/−^ MSNs by CD40-Fc, although on its own, Ani reduced neurite growth from *Cd40*^−/−^ MSNs as effectively as CD40-Fc (Figure 2d,e). Taken together, these findings suggest that JNK activity is necessary for the enhanced neurite growth phenotype of *Cd40*^−/−^ MSNs.

ERK1/ERK2 was activated by the treatment with the flavonoid Fisetin (Fis) [20,21]. Treatment with Fis promoted a significant but partial reduction in neurite growth (white bar), but it significantly inhibited the rescue of the exuberant phenotype of *Cd40*^−/−^ MSNs by CD40-Fc (grey bar) (Figure 2f,g). The activation of ERK1/ERK2 was prevented by treating the neurons with U0126, a selective MEK1/MEK2 inhibitor that interferes with MEK1/MEK2-dependent activation of ERK1/ERK2. U0126 did not prevent the rescue of the enhanced neurite growth phenotype of *Cd40*^−/−^ MSNs by CD40-Fc (Figure 2f,g). No significant difference was observed in total neurite length of *Cd40*^−/−^ MSNs treated with CD40-Fc and MSNs treated with CD40-Fc plus U0126 (Figure 2g). The treatment of *Cd40*^−/−^ MSNs with Fc plus U0126 did significantly reduce neurite growth, though not as effectively as CD40-Fc alone or CD40-Fc plus U0126 (Figure 2g). This suggests that without CD40L reverse signalling, there is some basal activation of ERK1/ERK2 that favours the exuberant growth of MSN neurites. The analogue inactive control of U0126, U0124, had no significant effect on neurite length (not shown). Taken together, these results suggest that ERK1/ERK2 activation does not contribute to the reduced neurite growth response of MSNs to CD40L reverse signalling, although manipulation of ERK1/ERK2 signalling can affect MSN neurite growth.

In a previously published study, we observed that CD40L reverse signalling, in addition to regulating the length of neurites, also modulates the number of branches points per neurite without affecting significantly the number of neurites emerging from the soma [1]. In this study, we also evaluated the effect of these pharmacological reagents on the number of ramifications per neurite. As shown in Appendix A, the manipulation of these signalling pathways produces a similar effect on the number of branches per neurite as in the neurite growth, suggesting that CD40L reverse signalling also influences neurite branching. Because the regulation of neurite branching follows a similar pattern as neurite growth for the rest of the study, we focused on neurite growth.

Taken together, these findings suggest that PKC and JNK activity mediate the effect of CD40-activated CD40L reverse signalling on neurite growth suppression from developing MSNs, while ERK1/ERK2 has an opposite and modulated influence on neurite growth restraint.

### 3.3. Effects on Neurite Growth of Manipulating PKC, JNK, and ERK1/ERK2 Signalling Pathways in Combination

We used pharmacological activators (labelled in green) and inhibitors (labelled in red) of PKC, JNK, and ERK1/ERK2 in combination to determine their potential functional interactions in regulating the restraint of neurite growth by CD40L reverse signalling. In these experiments, we treated *Cd40*^−/−^ MSNs with CD40-Fc to activate CD40L reverse signalling and either activated or inhibited either PKC or JNK or ERK1/ERK2 either alone or together with pharmacological manipulation (either activation or inactivation) of the remaining two signalling pathways. To simplify graphical presentation of the key data, only the combinations where the other pharmacological reagent has the opposite effect on growth are shown in Figure 3. For instance, CD40-Fc-treated neurons were stimulated with the activator of PKC, which restrains neurite growth when CD40L reverse signalling is activated, either alone or in combination with either the inhibitor of JNK or the activator of ERK1/ERK2 that reverse the effect of CD40L reverse signalling on growth restriction. For completeness, all combinations are shown in Appendix A. In these experiments, MSN cultures were treated 24 h after plating with the combined reagents and analyzed after 10 days in vitro. Individual treatments with either CD40-Fc or Fc were measured for comparison.

As shown in Figure 2, maximal depression of neurite growth by CD40L reverse signalling requires activation of PKC and JNK and the inhibition of ERK1/ERK2. When PKC was activated, the extent of neurite growth did not revert to the extent observed in Fc-treated neurons by manipulating either of the other two pathways (either inhibiting JNK or activating ERK1/ERK2) (Figure 3a). Similarly, when JNK was activated, neither inhibition of PKC nor activation of ERK1/ERK2 affected the effect of JNK activation on suppression of neurite growth (Figure 3b). These results suggest that when CD40L reverse signalling is activated, the regulatory function on neurite growth of either PKC activation or JNK activation are not regulated by negative manipulation of either of the other two signalling pathways. As already shown, treatment with the PKC inhibitor prevented the inhibition of neurite growth by CD40L reverse signalling (Figure 3a). This prevention of neurite growth inhibition by PKC activation was prevented by either simultaneous activation of JNK or by simultaneous inhibition of ERK1/ERK2 (Figure 3a). As already shown, inhibition of neurite growth by CD40L reverse signalling was prevented when JNK was inhibited (Figure 3b). However, the concomitant activation of PKC was able to significantly affect the effect of JNK inhibition (Figure 3b), although concomitant inhibition of ERK1/ERK2 did not affect the effect of JNK inactivation on growth (Figure 3b). This suggests that the inhibition of ERK1/ERK2 when JNK is inhibited is not sufficient to affect neurite growth.

ERK1/ERK2 activation reduced the ability of CD40L reverse signalling to restrain neurite growth (Figure 3c). However, in agreement with the predominant role of activation of PKC and JNK in mediating neurite growth inhibition in response to CD40L reverse signalling, the concomitant activation of either PKC or JNK cancelled out the effect of activation of ERK1/ERK2 alone (Figure 3c). In the case of the combined activation of JNK and ERK1/ERK2, the restriction of neurite growing was even more effective than with CD40-Fc alone (Figure 3c and Appendix A). Inhibition of ERK1/ERK2 did not affect growth inhibition by CD40L reverse signalling and was not further affected by concomitant inhibition of PKC. However, the concomitant inhibition of JNK was able to partially but significantly affect neurite growth (Figure 3c).

Activation of either JNK or PKC is required for suppression of neurite growth by activation of CD40L reverse signalling. The manipulation of the other two signalling pathways to suppress the neurite overgrowth is ineffective when PKC or JNK are activated (Figure 3a,b and Appendix A). Interestingly, the activation of JNK even produces a more restricted growth than CD40-Fc alone (Appendix A). The inhibition of JNK that suppresses the control over neurite growth is only restored by activation of PKC (Figure 3b and Appendix A). However, the inhibition of neurite growth by PKC activation is restored by both activating JNK and inhibiting ERK1/ERK2 (Figure 3a and Appendix A). ERK1/ERK2 activation is unable to prevent the growth inhibitory effect of CD40L reverse signalling when JNK and PKC are simultaneously inhibited (Appendix A). As expected from the dominant role of JNK or PKC over the control of neurite growth, the prevention of CD40-Fc-induced neurite growth suppression by activation of ERK1/ERK2 is inhibited by activation of either JNK or PKC (Figure 3c and Appendix A). While these results suggest a more dominant role for JNK and PKC activation on CD40-Fc-induced neurite growth inhibition, ERK1/ERK2 plays a modulatory role. Taken together, our data suggest that JNK, PKC, and ERK1/ERK2 signalling pathways participate in an interacting network to regulate neurite growth suppression from MSNs by CD40L reverse signalling.

### 3.4. Effect of Pharmacological Manipulation on Phosphorylation of Signalling Pathways

By means of western blotting experiments, we determined the degree of phosphorylation of JNK, PKC, and ERK1/ERK2 following the pharmacological manipulations of the other two signalling pathways; i.e., the effect on phospho-JNK of PKC and ERK1/ERK2 manipulation individually and in combination. With these experiments, we aimed to establish the effect on activity of these proteins of the pharmacological manipulation of either one or both signalling pathways. Striatal MSNs from E14 embryos of *CD40*^−/−^ mice plated with complete medium were treated after 9 days in vitro with either Fc or CD40-Fc without and with pharmacological reagents individually or in combination for 20 min, the time when all three phospho-proteins reach their respective peaks (Figure 1). After treatments, neuronal lysates were probed for either phospho-JNK, phospho-PKC, or phospho-ERK1/phospho-ERK2. Representative western blots probed for phospho-JNK (Figure 4a), phospho-PKC (Figure 4b), and phospho-ERK1/phospho-ERK2 (Figure 4c) are shown. To study the effect on phosphorylation when CD40L reverse signalling was activated, bar charts show the quantification of the relative densitometries from multiple blots for phospho-JNK (Figure 4d), phospho-PKC, (Figure 4e) and phospho-ERK1/phospho-ERK2 (Figure 4f) for the neurons treated as indicated in the presence of CD40-Fc (grey bars). The quantification of the basal effect (with Fc) and the differences in phosphorylation levels for a particular reagent in the presence of Fc or CD40-Fc are shown in Appendix A.

Confirming our previous results, there were significant increases in the levels of all three phospho-proteins after CD40-Fc treatment compared with Fc treatment after 20 min (Figure 4a–f). In the presence of CD40L reverse signalling, a small, statistically insignificant reduction of the phospho-JNK level compared with CD40-Fc was detected when ERK1/ERK2 was manipulated (Figure 4d). In the case of PKC, only its activation caused a significant reduction in the phospho-JNK level compared to CD40-Fc treatment alone (Figure 4d). Interestingly, no significant differences compared to CD40-Fc were detected in the combined treatment with PKC activator and ERK1/ERK2 inhibitor (Figure 4d) or with the combined treatment with PKC inhibitor and ERK1/ERK2 activator (Figure 4d). However, in the presence of CD40L reverse signalling, the combined treatment with the activators of PKC and ERK1/ERK2 significantly increased the levels of phospho-JNK, while the individual activation of PKC or ERK1/ERK2 produced low levels of phospho-JNK (Figure 4d). Likewise, combined treatment with the inhibitors of PKC and ERK1/ERK2, that individually did not have any significant effect on the phospho-JNK level, produced a significantly increased level of phospho-JNK (Figure 4d). These results indicate that when CD40L reverse signalling is activated, the level of phospho-JNK is differentially regulated depending on either combined activation or inhibition of either PKC and/or ERK.

In the regulation of the level of phospho-PKC, manipulation of ERK1/ERK2 produced opposite effects on the level of phospho-PKC. While the inhibition of ERK1/ERK2 drastically increased phospho-PKC, its activation significantly reduced phospho-PKC compared with CD40-Fc (Figure 4e). When JNK was manipulated, only its activation significantly increased phospho-PKC. In the presence of CD40L reverse signalling, independently of whether JNK was activated or inhibited, all combinations where ERK1/ERK2 were inhibited, the final level of phospho-PKC increased significantly compared to CD40-Fc (Figure 4e). However, in the combinations in which ERK1/ERK2 was activated (which reduced pPKC level on its own) the effect produced was reversed by either activation or inhibition of JNK, although the effect of the inhibitor of JNK was not statistically significant compared to CD40-Fc (Figure 4e). Taken together, these results show that when CD40L reverse signalling is activated, the level of phospho-PKC is regulated by both JNK and ERK1/ERK2, although to a greater extent than regulation of phospho-JNK by PKC and ERK1/ERK2.

In the presence of CD40L reverse signalling, manipulation of PKC produced opposite effects on the phospho-ERK1/phospho-ERK2 level. While the activator of PKC produced elevated phosphorylation of ERK1/ERK2, inhibition of PKC significantly reduced the levels of phospho-ERK1/phospho-ERK2 (Figure 4f). The inhibition of JNK produced similar levels of phospho-ERK1/phospho-ERK2 as CD40-Fc alone, and its activation increased more than CD40-Fc (Figure 4f). The elevated levels of phospho-ERK1/phospho-ERK2 observed with the individual treatments with PKC or JNK activators were maintained in the combined treatment (Figure 4f). However, the increase in phospho-ERK1/phospho-ERK2 with the activator of JNK was drastically reduced when JNK was activated simultaneously with the inhibition of PKC (Figure 4f). Independently of the activation or inhibition of JNK, the inhibition of PKC produced a significant reduction in the phospho-ERK1/phospho-ERK2 (Figure 4f). These results show that when CD40L reverse signalling is activated, the level of phospho-ERK1/phospho-ERK2 is regulated by both JNK and PKC, and the effect on the levels of phospho-ERK1/phospho-ERK2, is differentially modulated depending on either combined activation or inhibition of either PKC and/or JNK.

Taken together, the above findings suggest that JNK, PKC, and ERK1/ERK2 signalling pathways regulate the growth of neural processes by functioning as an interacting regulatory network in which these signalling pathways influence and modulate the activity of one another with distinctive final effects. However, these three signalling pathways influence neurite growth to different extents. The levels of phospho-PKC and phospho-ERK1/phospho-ERK2 are more influenced by the other two signalling pathways than the level of phospho-JNK.

### 3.5. Regulation of MSN Neurite Growth by CD40L Reverse Signalling Is Mediated by an Interacting Network Comprised of JNK, PKC, and ERK1/ERK2

The results presented above suggest that neural growth from MSNs is regulated by an interacting network comprised of at least three well-established signalling pathways. The interaction between JNK, PKC, and ERK1/ERK2 and their consequences on growth are summarized in Figure 5.

Activation of CD40L reverse signalling enhances the phosphorylation and hence activation of JNK, PKC, and ERK1/ERK2. Activation of JNK increases the phosphorylation of PKC with the concomitant restrain in growth independent of the activation state of ERK1/ERK2. However, when the inhibitor of JNK, which does not have any effect over the phosphorylation of PKC, is added simultaneously with the inhibitor of ERK1/ERK2, the level of phospho-PKC increases, and depending on the final balance of PKC activation, neurite overgrowth will or will not be restrained (see in Appendix A: CD40-Fc + JNK + ERK + PKC vs. CD40-Fc + JNK + ERK + PKC). By contrast, the activator of PKC negatively modulates phospho-JNK, but the simultaneous addition of the activator of ERK1/ERK2 increases phospho-JNK with the restriction of neurite overgrowth. As occurred with PKC, depending on the final level of JNK activation, the simultaneous inhibition of PKC and ERK restrains or not the neurite overgrowth (see in Appendix A: CD40-Fc + PKC + ERK + JNK vs. CD40-Fc + PKC + ERK + JNK). The phosphorylation of PKC and ERK are reciprocally regulated. The activator of PKC increases phospho-ERK1/phospho-ERK2 and the inhibitor of PKC reduces phospho-ERK1/phospho-ERK2, while the activator of ERK1/ERK2 reduces phospho-PKC and the inhibitor of ERK1/ERK2 increases phospho-PKC. Activation of JNK also modulates positively the level of phospho-ERK1/phospho-ERK2. The simultaneous activation of JNK and PKC increases phospho-ERK1/phospho-ERK2. However, the inhibition of PKC that reduces phospho-ERK1/phospho-ERK2 is able to reduce the phosphorylation of ERK1/ERK2 even when JNK is simultaneously activated. Because in both situations, JNK is activated, independently on the effect on phospho-ERK1/phospho-ERK2 level, there is a restrain in the overgrowth. The simultaneous inhibition of JNK and PKC also reduces the level of phospho-ERK1/phospho-ERK2, but in this case, there is not restriction in the overgrowth. However, in contrast with JNK and PKC, this loss of restrain over the growth is not regulated by the final balance of ERK1/ERK2 activation.

Taken together, these findings suggest that, due to the restraint in the neurite growth brought about by CD40L reverse signalling, activation of JNK is decisive. Indeed, all conditions in which JNK was activated produced an even more restricted neurite growth. Moreover, the function of JNK is little modulated by the activation state of PKC or ERK1/ERK2. The activation state of PKC also influences, decisively, neurite outgrowth. However, in this case, the function of PKC is regulated by the other two signalling pathways. Finally, ERK1/ERK2 appears to have a modulator role, especially over the function of PKC. Interestingly, the inhibition of ERK1/ERK2 is sufficient to prevent the overgrowth of MSN neurites even when PKC is inhibited as long as JNK is active. As with PKC activity, the activation of ERK1/ERK2 is also highly regulated by the other two signalling pathways.

### 3.6. The Syk Tyrosine Kinase Is Expressed in Striatal Medium Spiny Neurons

In pyramidal neurons, activation of CD40L reverse signalling leads to recruitment of the protein Syk tyrosine kinase to the membrane together with PKCβ [2]. The protein Syk (spleen tyrosine kinase) is a non-receptor tyrosine kinase whose function has been extensively studied in the immune system [22,23]. In addition to the immune cells, recent work has shown that Syk mediates a variety of diverse biological functions in several different cell types after its activation and recruitment to the cell membrane (reviewed in [24]).

Initially, we analyzed the expression of Syk by western blotting in striatal lysates from *Cd40*^+/+^ and *Cd40*^−/−^ mice over a range of ages, from embryonic E14 to adult (Figure 6a). Full-length Syk is a 72 kDa protein that is proteolytically cleaved to generate a 40 kDa protein fragment that contains the catalytic domain [25]. Both the 72 kDa and 40 kDa isoforms were detected in striatal lysates from *Cd40*^+/+^ and *Cd40*^−/−^ mice. The pattern of the 40 kDa catalytic fragment was comparable between *Cd40*^+/+^ and *Cd40*^−/−^ mice. With few variations, the expression was similar over the range of ages analyzed (Figure 6a). In both cases, expression of the 72 kDa protein Syk began during the perinatal period and increased with the age. The expression pattern of the 40 kDa and 72 kDa isoforms observed in vivo was replicated in lysates of neurons cultured from *Cd40*^−/−^ mice with time in vitro (Figure 6b).

We also analyzed the expression of PKCβ (the main isoform that mediates the neurite growth-promoting effects of CD40L reverse signalling in hippocampal pyramidal neurons), PKCγ (the isoform that mediates the neurite growth-restraining effects of CD40L reverse signalling in MSNs), and CD40L. No clear differences in the expression of these two PKC isoforms were observed between lysates from *Cd40*^+/+^ and *Cd40*^−/−^ mice (Figure 6a). The expression of the two isoforms of PKCβ (βI/βII) started to be clearly expressed from E18 onward (Figure 6a). Interestingly, the expression of PKCγ started at P1 and its expression increased markedly with the age thereafter (Figure 6a). As observed in pyramidal neurons [2], the expression of CD40L showed some differences between *Cd40*^+/+^ and *Cd40*^−/−^. In wild-type mice, the expression began earlier and appeared more sustained during early life of the animals, while the expression in CD40-null animals was slightly delayed and more robust (Figure 6a). In cultured neurons from *Cd40*^−/−^ mice, the expression of PKCβ, PKCγ, and CD40L was similar to the expression observed in striatal lysates from CD40-null mice (Figure 6b).

### 3.7. CD40-Activated CD40L Reverse Signalling Recruits Syk but Not PKCγ

We conducted immunoprecipitation experiments to identify membrane-associated molecular components of the activated CD40L receptor complex. For these experiments, we cultured E14 striatal MSNs from *Cd40*^−/−^ for 9 days. Neurons were treated with either Fc or CD40-Fc for 30 min. After pulling down the Fc fragment from the neuronal lysates with protein G-Sepharose, we used western blotting to detect the physical interaction of CD40L, Syk, PKCβ, and PKCγ (Figure 7a). Quantification of the associated proteins after immunoprecipitation is shown in Figure 7b. As expected, after stimulation of CD40L reverse signalling, we detected the association of CD40L in cultures treated with CD40-Fc but not in cultures treated with Fc (Figure 7a,b). Both the 72 kDa and 40 kDa Syk were expressed at 9 days in vitro. After immunoprecipitation (IP), a slight interaction was detected with the 72 kDa isoform in the CD40-Fc-treated cultures (Figure 7a), but no statistical differences were detected between Fc and CD40-Fc (Figure 7b). The interaction of the 40 kDa Syk catalytic fragment was clearly detected after treating the neurons with CD40-Fc, showing a statistical difference compared with neurons treated with Fc (Figure 7a,b). Although both PKC isoforms were clearly detectable in the lysates prior to G-Sepharose pull down, neither of these PKC isoforms (PKCβ or PKCγ) was detected after IP (Figure 7a,b).

These results indicate that activation of CD40L reverse signalling in striatal MSNs promotes the recruitment of Syk, in particular, the 40 kDa catalytic fragment. However, in contrast to excitatory pyramidal neurons, PKC does not form part of this signalling complex. This agrees with the fact that PKCγ, the main isoform of PKC that mediates the morphological effects of CD40L reverse signalling in MSNs, does not interact with Syk, as does PKCβ. In silico determination using the STRING database (http://string-db.org/ (accessed on 12 March 2021)) of all possible protein-protein interactions (based on experiments, co-expression, co-occurrence, gene fusion, neighbourhood, databases, and text-mining) between mouse CD40L and PKCγ showed that this PKC isoform does not interact with Syk (Appendix A). However, CD40L and PKCγ interact with some common partners that include members of JNK and ERK and associated regulatory proteins of these signalling pathways (Appendix A).

## 4. Discussion and Conclusions

Both CD40L-activated CD40 forward signalling and CD40-activated CD40L reverse signalling are relevant physiological regulators of growth and elaboration of neural processes in several population of neurons over different developmental periods [1,12,13,26,27]. Distinctive cellular responses are activated by reverse or forward signalling [28,29]. In the developing CNS, CD40-activated CD40L reverse signalling has opposite effects on neurite growth from excitatory hippocampal pyramidal neurons and inhibitory striatal MSNs. In pyramidal neurons, CD40L reverse signalling enhances dendrite and axon growth, while in MSNs, it represses neurite growth [1]. Using a variety of pharmacological reagents, we previously revealed that in pyramidal neurons, PKC, ERK1/ERK2, and JNK signalling pathways act as an interconnected and interdependent signalling network to mediate effects of CD40L reverse signalling [2]. Here we have demonstrated that these three signalling pathways also participate in an interconnected and interdependent signalling network, but their interdependencies and interconnections differ from those in pyramidal neurons.

Western blot studies revealed that CD40-activated CD40L reverse signalling promoted the phosphorylation, and therefore the activation, of PKC, JNK, and ERK1/ERK2. Treatment with either activators or inhibitors of PKC, JNK, and ERK1/ERK2 showed that these signalling proteins regulate neurite growth responses to CD40L reverse signalling in distinctive ways. In MSNs, activators of PKC and JNK produced a similar effect to CD40L and did not significantly affect CD40L-restrained growth. Accordingly, inhibitors of either PKC or JNK eliminated the control exercised by CD40L reverse signalling but had no effect in the absence of CD40L reverse signalling. The effects observed using activators or inhibitors of ERK1/ERK2 showed that the function of this signalling pathway is opposite to PKC and JNK. In the presence of CD40L reverse signalling, the activator of ERK1/ERK2 eliminated the effect of CD40L, but the inhibitor of this signalling pathway did not affect the influence of CD40L reverse signalling. These findings suggest that PKC activation and JNK activation are involved in mediating the effects of CD40L reverse signalling in MSNs, whereas ERK1/ERK2 activation is not. Taken together, our results suggest that although the signalling pathways involved in mediating the opposite effects of CD40L reverse signalling on neurite growth from pyramidal and MSNs are the same (JNK, PKC, and ERK1/ERK2), they are involved in distinctive ways. In contrast to MSNs, the growth-promoting of CD40L reverse signalling on pyramidal neurons requires inhibition of JNK together with the activation of PKC and ERK1/ERK2 [2].

Our studies, using a combination of pharmacological reagents, besides revealing the importance and potential functional interaction within the signalling network, also reveal similarities and differences in the regulation of neurite growth in response to CD40L reverse signalling in excitatory pyramidal neurons and inhibitory striatal MSNs. In pyramidal neurons, JNK plays a dominant role, its inhibition being essential to allow the growth-promoting actions of PKC and ERK1/ERK2 [2]. In MSNs, JNK has a predominant role, its activation being required for the restriction of neurite overgrowth, and only activation of PKC restores control over neurite overgrowth when JNK is inhibited. Indeed, in *Cd40*^−/−^ MSNs, the activator of JNK produces shorter neurites than CD40-Fc alone. The data raise the possibility that the primary function of JNK is related to determining the extension of growing processes. In pyramidal neurons, where CD40L reverse signalling promotes axon and dendrite growth, JNK controls the termination of the growth response and in MSNs, where CD40L reverse signalling restrains neurite growth, it controls excessive growth.

The activation of PKC is a common requirement in the morphological effects on neural processes brought about by CD40L reverse signalling in both neurons. However, while in pyramidal neurons, its function is not regulated by ERK1/ERK2, in MSNs, PKC function is modulated by ERK1/ERK2. In both kinds of neurons, ERK1/ERK2 seems to play a modulatory role whose function is regulated by the other two signalling pathways in the network. However, in pyramidal neurons, its activation participates in mediating the effect of CD40L on axon and dendrite growth, while in MSNs, its inhibition is required for the effect of CD40L on neurite growth. Thus, ERK1/ERK2 allows growth when activated and restricts growth when inhibited.

The phosphorylation studies show that the degree of phosphorylation, and consequently the activation, of a particular signalling protein is modulated by the other two signalling proteins in the interacting regulatory network. In the presence of CD40L reverse signalling, the level of phospho-JNK is less influenced than phospho-PKC and phospho-ERK1/phospho-ERK2 by manipulating the other two signalling pathways in the regulatory network either individually or in combination using activators or inhibitors. This highlights another difference between MSNs and pyramidal neurons. In the presence of CD40L reverse signalling, in pyramidal neurons, the level of phospho-JNK is more influenced by the other two signalling pathways, both individually or in combination, than the influence of the other two signalling pathways on phospho-PKC or phospho-ERK1/phospho-ERK2 [2].

We reported that the intracellular signalling initiated downstream CD40L in pyramidal neurons involves recruitment of the protein Syk to the membrane. Because CD40L, like other members of the tumor necrosis factor superfamily (TNFSF), lacks enzymatic activity, it needs to recruit adaptor and effector proteins to initiate reverse signalling [28]. Syk functions not only as a tyrosine kinase that phosphorylates downstream substrates, but also as an adaptor protein with the ability to bind to diverse signalling proteins. The signalling outcome depends on the particular proteins with which Syk interacts, for example, cell proliferation or differentiation in the case of B cells [23]. We demonstrated by in silico analysis and immunoprecipitation studies that CD40L, PKCβ, and Syk form a receptor complex following activation of CD40L reverse signalling in hippocampal pyramidal neurons [2]. Syk is also expressed in MSNs, although the expression pattern shows some differences from that observed in hippocampal neurons. In hippocampal pyramidal neurons, the 40 kDa catalytic fragment is expressed during perinatal age, and from postnatal 6 days (P6), decreases with age to low, undetectable levels in adults [2]. In MSNs, the expression level of the 40 kDa isoform is fairly constant from E14 to adult. The expression of the 72 kDa full-length isoform is similar in both kinds of neurons. It is undetectable until P3 in hippocampal neurons and until P1 in MSNs, after which it increases to achieve the highest level in the adult. Immunoprecipitation studies showed that following activation of CD40L reverse signalling in MSNs, the 40 kDa catalytic Syk interacts with CD40L, but we did not detect interaction with either PKCγ or PKCβ.

PKC proteins are serine/threonine kinases that are classified into three subfamilies (classical or conventional, novel, and atypical) depending on their second messenger requirements. They play key roles in signalling pathways that regulate a variety of cellular functions, including regulating the growth and branching of neural processes [1,10,30,31,32,33,34]. By activating different PKC isoforms, CD40L reverse signalling regulates both the growth-promoting and the growth-inhibitory effects in different kinds of neurons [1]. The intracellular signalling complex initiated downstream CD40L in pyramidal neurons is different from the signalling complex in MSNs. In striatal GABAergic MSNs, activation of CD40L reverse signalling is followed by the recruitment of Syk, and although PKCγ is involved in the morphological effects of CD40L reverse signalling, PKCγ does not form part of the initial signalling complex. PKCγ is also involved in the development of Purkinje cells [35,36], which are GABAergic neurons located in the cerebellum. PKCγ functions as a negative regulator of dendritic growth and branching in these GABAergic neurons [37,38]. PKCγ-deficient mice have altered dendritic Purkinje cell development with dendritic trees larger and with more branching points compared with wild-type Purkinje neurons. The activation of PKC with PMA reduces the dendritic arbors restoring the wild-type phenotype, while PKC inhibition does not produce any significant morphological change [38].

CD40L reverse signalling promotes axon and dendrite growth in pyramidal neurons and restrains neurite growth in MSNs. These different outcomes result from the participation of the same elements. Both neuron types also express CD40L, PKCβ, PKCγ, and Syk. However, differences in temporal window expression, protein levels, and different protein context in each kind of neuron produce a different initial signalling complex following activation of CD40L reverse signalling. This difference in the initial signalling complex might explain the differences in the phosphorylation kinetics observed in both kind of neurons after activation of CD40L reverse signalling in absence of pharmacological reagents. Because the control of neural process growth evolves over time, these differences might influence the temporal interaction and participation of other factors. It should be noted that in spite of the described specificity of the activators and inhibitors used in this study, as with all pharmacological reagents, off-target effects may occur. Thus, while additional studies using alternative approaches may be beneficial, our work has begun to characterize the role and potential interactions of JNK, PKC, and ERK1/ERK2 in the control of neural process growth following CD40L reverse signalling.

## Figures and Tables

**Figure 1 cells-10-00829-f001:**
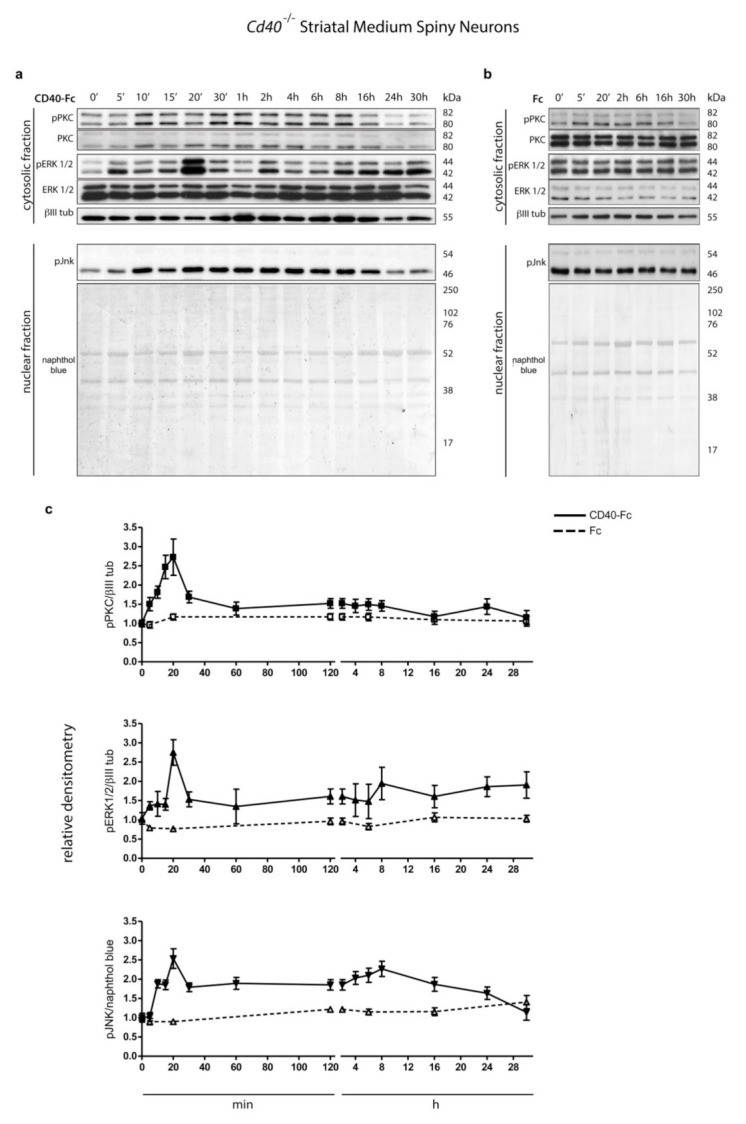
Protein kinase C (PKC), extracellular regulated kinases 1 and 2 (ERK1/2), and c-Jun *N*-terminal kinase (JNK) phosphorylation after stimulating CD40L reverse signalling. (**a**,**b**) Representative western blots of lysates of *Cd40*^−/−^ E14 striatal medium spiny neuron (MSN) cultures treated for the indicated times with (**a**) 1 μg/mL CD40-Fc or (**b**) 1 μg/mL Fc protein as a control. Lysates were prepared from all cultures after a total of 9 days in vitro. The blots were labelled with anti-phopho-PKC^Thr514^ (pPKC), anti-phospho-p44/p42^Thr202/Tyr204^ MAPK (ERK1/2) (pERK 1/2), anti-phospho-SAPK/JNK^Thr183/Tyr185^ (pJNK). Anti-PKC (PKC), anti-p44/p42 (ERK1/2), and anti-βIII tubulin (βIII tub) were used as loading control for the cytosolic fractions and naphthol blue was used as loading control in the nuclear fraction. (**c**) Densitometry of at least three independent western blots using βIII tubulin (βIII tub) for normalising pPKC and pERK1/2 and naphthol blue for pJNK (mean ± s.e.m.).

**Figure 2 cells-10-00829-f002:**
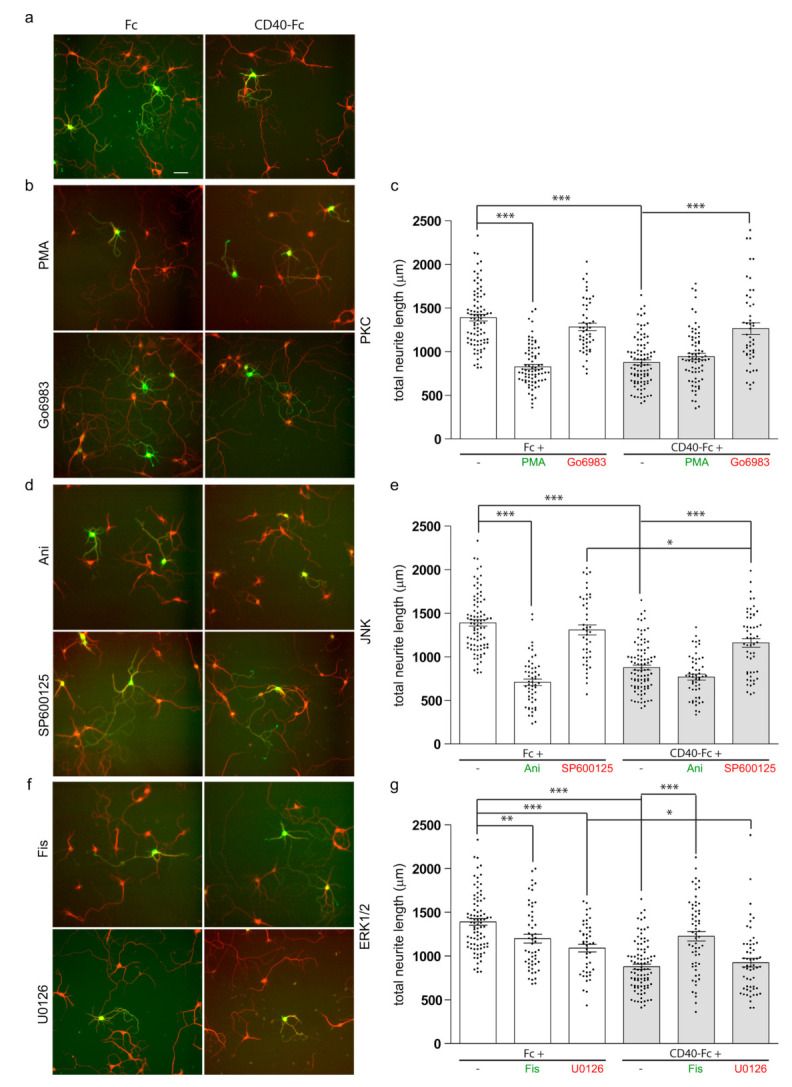
Effect of pharmacological reagents on neurite growth from MSNs. (**a**,**b**,**d**,**f**) Representative photomicrographs of Fc-treated and CD40-Fc-treated MSNs exposed to pharmacological reagents. (**c**,**e**,**g**) Quantification of the influence on total neurite length of pharmacological reagents. Cultures of striatal MSNs were established from E14 *Cd40*^−/−^ embryos. The cultures were treated 24 h after plating with either 1 μg/mL of Fc (white bars) or 1 μg/mL of CD40-Fc (grey bars) together with pharmacological manipulators of PKC (either 500 nM phorbol-12-myristate-13-acetate (PMA) or 500 nM Go6983) (**b**,**c**), JNK (either 50 nM Ani or 1 μM SP600125) (**d**,**e**), and ERK1/ERK2 (either 1 μM Fisetin (Fis) or 1 μM U0126) (**f**,**g**). The neurons were double labelled for βIII tubulin (red) and DARPP-32 (green) to identify MSNs after a total of 10 days in vitro. Scale bar, 50 μm. For clarity, the activators are labelled in green and the inhibitors in red. In the scatter charts, mean ± s.e.m of at least three independent experiments are shown. The dots represent the data obtained from individual neurons (number of neurons per condition are given in Appendix A). One-way ANOVA with multiple Newman–Keuls statistical comparison. Key statistical significance differences are indicated (*** *p* < 0.001, ** *p* < 0.01, and * *p* < 0.05). Comprehensive statistical analysis is provided in the Appendix A.

**Figure 3 cells-10-00829-f003:**
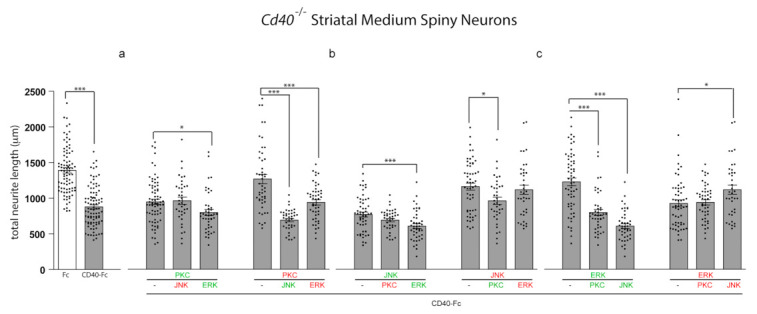
The influence of pharmacological reagents in combination on neurite growth from MSNs. (**a**–**c**) Scatter charts of total neurite lengths of MSNs of E14 *Cd40*^−/−^ embryos cultured for 10 days in vitro and treated 24 h after plating with 1 μg/mL CD40-Fc (grey bars) plus either activators (green) or inhibitors (red) of (**a**) PKC, (**b**) JNK, or (**c**) ERK1/ERK2 in combination with the activators or inhibitors of the other two pathways (same concentrations were used as in Figure 2). For comparison, neurite lengths of neurons in cultures treated with 1 μg/mL control Fc alone (white bars) are shown. The mean ± s.e.m of at least three independent experiments are shown. The dots represent the data obtained from individual neurons (number of neurons per condition in Appendix A). One-way ANOVA with multiple Newman–Keuls statistical comparison, *** *p* < 0.001 and * *p* < 0.05.

**Figure 4 cells-10-00829-f004:**
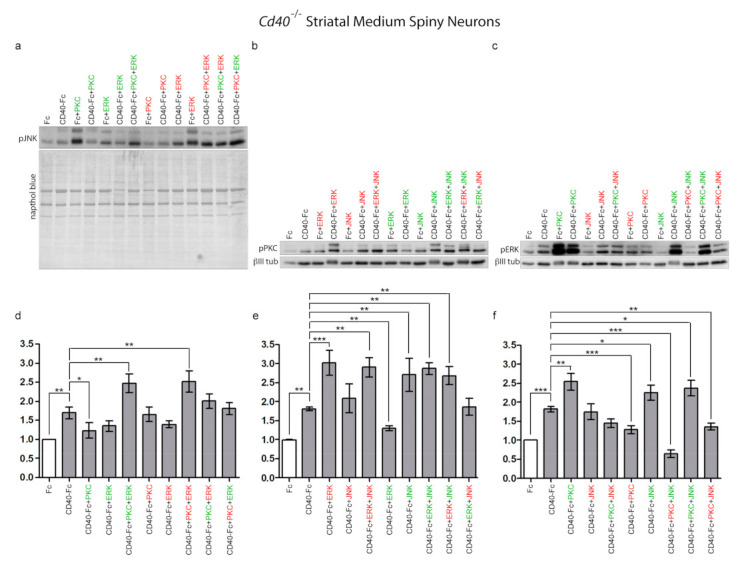
Regulation of JNK, PKC, and ERK1/ERK2 phosphorylation by pharmacological reagents in MSNs in the presence and absence of CD40L reverse signalling. (**a**–**c**) Representative western blots of lysates of MSNs of *Cd40*^−/−^ E14 embryos cultured for 9 days and treated for 20 min with either 1 μg/mL control Fc or 1 μg/mL CD40-Fc in combination with activators (green) and inhibitors (red) of PKC, JNK, and ERK as indicated. The concentrations were the same as those indicated in Figure 2. The western blots were probed with anti-pJNK after treatment with activators and inhibitors of PKC and ERK1/ERK2 (**a**), anti-pPKC after treatment with activators and inhibitors of ERK1/ERK2 and JNK (**b**), and anti-pERK1/pERK2 after treatment with activators and inhibitors of PKC and JNK (**c**). Anti-βIII tubulin was used to normalize western blots of cytosolic fractions and Naphthol blue for the nuclear fractions. (**d**–**f**) Quantification of at least three independent western blots. The grey bars show combined treatments in the presence of 1 μg/mL CD40-Fc, and the white bar, the control with 1 μg/mL Fc. The mean ± s.e.m are indicated (*** *p* < 0.001, ** *p* < 0.01, and * *p* < 0.05, one-way ANOVA with multiple Newman–Keuls statistical comparison).

**Figure 5 cells-10-00829-f005:**
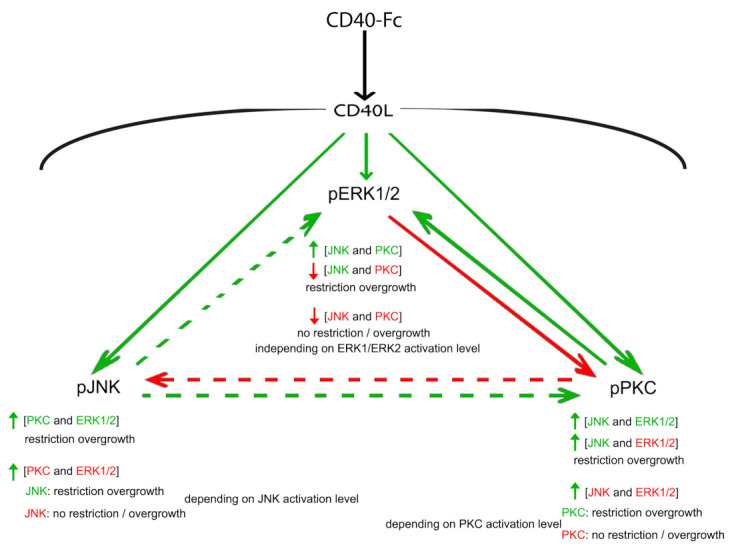
Schematic summary. Graphic summary of the interactions between the JNK, PKC, and ERK1/ERK2 downstream of the activation of CD40L reverse signalling and the effect on neural process growth. The direction of influence is indicated by the arrows. Solid arrows indicate the influence of activation that is opposite to the influence of inhibition (i.e., activator of ERK1/ERK2 reduces phospho-PKC and the inhibitor of ERK1/ERK2 increases phospho-PKC; or activator of PKC increases phospho-ERK1/phospho-ERK2 and the inhibitor of PKC reduces phospho-ERK1/phospho-ERK2); dashed arrows indicate the influence of activation that do not have any influence when inhibiting (i.e., activator of PKC reduces phospho-JNK, but the inhibitor does not have any effect). The change in the level of phosphorylation is indicated by colour (green denotes increased phosphorylation and red denotes decreased phosphorylation).

**Figure 6 cells-10-00829-f006:**
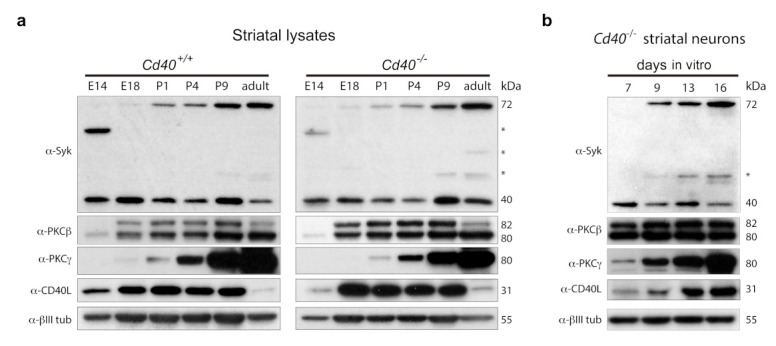
Expression of Syk, PKCβ, PKCγ, and CD40L. (**a**) Representative western blot of the expression of Syk, PKCβ, PKCγ, and CD40L in striatal lysates from *Cd40*^+/+^ and *Cd40*^−/−^ at the indicated ages. (**b**) Representative western blot of the expression of Syk, PKCβ, PKCγ, and CD40L in *Cd40*^−/−^ striatal neurons cultured for the days indicated. Anti-βIII tubulin was used as a loading control. * indicates nonspecific bands.

**Figure 7 cells-10-00829-f007:**
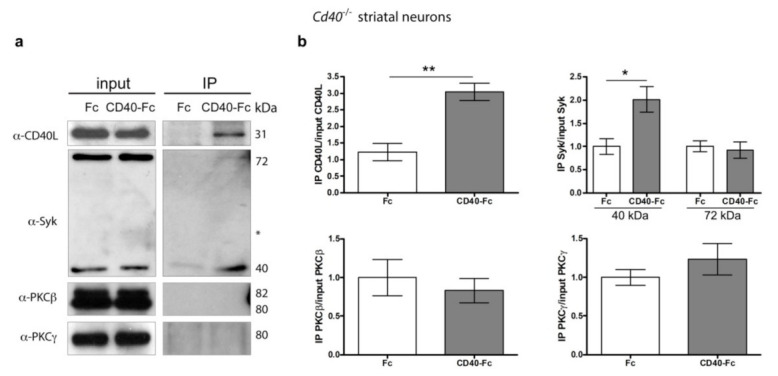
Pull down of Syk in *Cd40*^−/−^ striatal MSNs after CD40-activated CD40L reverse signalling. (**a**) Representative western blots of the expression of CD40L, Syk, PKCβ, and PKCγ in *Cd40*^−/−^ neurons from E14 embryos cultured for 9 days and treated for 30 min with either 1 μg/mL Fc or 1 μg/mL CD40-Fc, before pulled-down Fc fragment (input) and after pulled-down Fc fragment (IP). * = nonspecific band. (**b**) Quantification of at least three independent western blots of the quantity of CD40L, 40 kDa Syk, 72 kDa Syk, PKCβ, and PKCγ after IP normalizing to the total quantity of those proteins in the input. T-test: ** *p* < 0.01 and * *p* < 0.05.

## Data Availability

The data and datasets generated for this study are available on request to the corresponding author.

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
