# Peer review of "Signalling Pathways Mediating the Effects of CD40-Activated CD40L Reverse Signalling on Inhibitory Medium Spiny Neuron Neurite Growth"

_cells, 2021, doi:10.3390/cells10040829_

Round 1

Reviewer 1 Report

In this manuscript, P. Carriba and A. Davies explored how CD40-activated CD40L reverse signalling regulates dendrite growth of inhibitory GABAergic medium spiny dendrite neurons (MSNs). The study is an extension of previous reports from the authors focussing on CD40L reverse signalling in other types on neurons. The outcomes show similarities and differences in signalling components downstream of CD40L reverse signalling in MSNs, compared to those acting in other types of neurons.

A major limitation of this study is the lack of wild-type neurons, as all experiments were basically done only with CD40-/- cells, with the possibility of assessing CD40L reverse signalling and effects on a subfraction of cells established from mutant embryos. Nevertheless, placing the outcomes in this specific mutant context, which may be slightly different in wild type neurons, the overall study provides insights on how CD40L reverse signalling operates in MSNs.

The manuscript is well written, although the result section is a bit long and descriptive and can be shortened.

Minor comment: western blot data shown in Figure 4a are based on a rather strong variation of protein amounts loaded, as shown by napthol blue staining. It would be appropriate to show another western blot with adequate loading.

Author Response

In this manuscript, P. Carriba and A. Davies explored how CD40-activated CD40L reverse signalling regulates dendrite growth of inhibitory GABAergic medium spiny dendrite neurons (MSNs). The study is an extension of previous reports from the authors focussing on CD40L reverse signalling in other types on neurons. The outcomes show similarities and differences in signalling components downstream of CD40L reverse signalling in MSNs, compared to those acting in other types of neurons.

A major limitation of this study is the lack of wild-type neurons, as all experiments were basically done only with CD40-/- cells, with the possibility of assessing CD40L reverse signalling and effects on a subfraction of cells established from mutant embryos. Nevertheless, placing the outcomes in this specific mutant context, which may be slightly different in wild type neurons, the overall study provides insights on how CD40L reverse signalling operates in MSNs.

             The aim of our study was to unambiguously determine how CD40L reverse signalling regulates the neurite growth in MSNs. We used Cd40-/- neurons treated with CD40-Fc because only this treatment unequivocally activates reverse signalling and because the activation of CD40L reverse signalling in Cd40-/- neurons completely restores the wild-type phenotype. The results of studies using wild-type neurons would be confusing and difficult to interpret for several reasons. First, endogenous CD40L reverse signalling still occurs in wild-type neurons, and this may explain why treating wild-type neurons with CD40-Fc does not have any effect on neurite growth, as previously reported. Second, because both CD40L-mediated reverse signalling and CD40-mediated forward signalling occur in wild-type neurons, CD40-Fc and soluble CD40L (sCD40L) could have a variety of effects. For example, CD40-Fc could not only enhance reverse signalling but block forward signalling by competing with endogenous CD40, and sCD40L could not only enhance forward signalling but block reverse signalling by competing with endogenous CD40L. We edited the text to clarify these points.

             The manuscript is well written, although the result section is a bit long and descriptive and can be shortened.

             We have shortened and simplified the results section.

Minor comment: western blot data shown in Figure 4a are based on a rather strong variation of protein amounts loaded, as shown by napthol blue staining. It would be appropriate to show another western blot with adequate loading.

             We have replaced the western blot with the corresponding loading control of naphthol blue for another with less variation in the protein quantities loaded.

Reviewer 2 Report

The authors dissect the role of CD40L reverse signaling on dendrite growthrestriction in inhibitory striatal medium spiny neurons (MSNs) by analyzing  JNK, PKC and ERK activation/inhibition using pharmacological agents. Using a similar analysis approach as in their prior work on pyramidal neurons, the novelty of the paper lies in dissecting the differences in JNK/PKC/ERK pathways in MSN development compared to pyramidal neurons. They find JNK to be most important for growth restriction via CD40L reverse signaling, while all three kinases impact on the phosphorylation status of the other two and regulate their activation. Unlike in pyramidal neurons, PKC does not seem to be part of the Syk activation complex, which might be an explanation for opposite effects on dendrite growth.
Overall, the authors elucidate the dependencies of JNK, PKC and ERK on their activation status and phosphorylation in detail. This work highlights that growth control pathways of inhibitory neurons differ from excitatory neurons, which are important findings for a better understanding of overall network development. The work provides interesting insight into the developmental differences in  inhibitory neurons and their dependencies on CD40L reverse signalling. Thus I support publication of this work with some additional considerations.

Major points:

  1. The data is not presented in a very accessible way, which is in part by using the abbreviations of the various pharmacological reagents used. Due to the large number of comparisons and conditions it is very difficult to keep the overview.  Thus I would suggest simplifying the labeling by indicating the activation or inhibition of pathways (for instance ERK+/-) instead. Color coding might be another option to ease the reader through the complex figures, in particular Fig. 4, which is very hard to follow and reconcile.
  2. In Fig. 4 many comparisons are made, but I wonder if all of these are relevant for the main theme of the work, namely the elucidation of CD40L reverse signalling in restricting dendrite growth in MSNs. While the pharmacological effects on the phopho-status of the respective kinases with Fc only are important controls, the reported changes do not reflect CD40L reverse signalling effects and make it hard to keep an overview. I would suggest to focus on comparisons in the presence of Fc-CD40L, as this is relevant for dendrite restriction of MSNs and better reflects the potential reciprocal effects of JNK/PKC/ERK depending on their activation status. Fc controls could be shown as a supplement instead. This would also allow for easier comparison with the respective dendrite phenotypes shown in the supplement.
  3. Detailed statistics (including n numbers) are missing and should be added at least as a supplemental file. Otherwise it is not clear if the data without shown is significantly changed or not. For example in Fig. 2, does Anisomycin significantly reduce dendrite length with Fc-CD40 compared to control? Moreover, the control plots for Fc and CD40-Fc in Fig. 2.c,e,g show the same data points. This should be at least mentioned in the legend. Exact n numbers should be stated in the legend as well.
  4. The legend for Suppl. Fig. 1 is missing and statistics should be included there.
  5. In line 519 the authors state that “the function of JNK is little modulated by the activation state of PKC or ERK1/ERK2.” and conclude that activation of JNK is decisive for the described activity and dendrite growth restraint. However, PKC manipulation equally affects the pJNK status, and dendrite growth restriction with PMA is not strongly affected by JNK/ERK manipulation (no significances are given in Suppl. Fig,1). While I do not argue with the overall conclusion of the authors, clarification of this point would be helpful.

Minor  points:
1. In the initial part of the introduction references should be cited to support the authors’ statements (e.g. lines 39-44).
2. Typo in line 70 “CD40”-L
3. Fig. 2 does not show >60 dots. Please check and update statistics.
4. In lines 277-279 the authors claim that basal ERK activity favors growth, while its activation via Fis shows a reduction. Thus ERK pathway activity seems to somehow reduce growth in both cd40-/- conditions. The authors do not explain this effect. Is this due to PKC phosphorylation?
5. line 348:“The  inhibition of JNK that suppresses the neurite growth is only restored by activation of PKC” sounds as if JNK inhibition suppresses dendrite growth, which is not the case
6. Line 426 typo “effects” should be singular
7. Line 568 Typo “MNSs”
8. Fig 5: To make it more clear, the strength of phosphorylation might be indicated by the stroke width of the arrows. A color code might also help to follow the figure

Author Response

The authors dissect the role of CD40L reverse signaling on dendrite growthrestriction in inhibitory striatal medium spiny neurons (MSNs) by analyzing  JNK, PKC and ERK activation/inhibition using pharmacological agents. Using a similar analysis approach as in their prior work on pyramidal neurons, the novelty of the paper lies in dissecting the differences in JNK/PKC/ERK pathways in MSN development compared to pyramidal neurons. They find JNK to be most important for growth restriction via CD40L reverse signaling, while all three kinases impact on the phosphorylation status of the other two and regulate their activation. Unlike in pyramidal neurons, PKC does not seem to be part of the Syk activation complex, which might be an explanation for opposite effects on dendrite growth.
Overall, the authors elucidate the dependencies of JNK, PKC and ERK on their activation status and phosphorylation in detail. This work highlights that growth control pathways of inhibitory neurons differ from excitatory neurons, which are important findings for a better understanding of overall network development. The work provides interesting insight into the developmental differences in  inhibitory neurons and their dependencies on CD40L reverse signalling. Thus I support publication of this work with some additional considerations.

             The reviewer correctly highlights our findings.

Major points:

            The data is not presented in a very accessible way, which is in part by using the abbreviations of the various pharmacological reagents used. Due to the large number of comparisons and conditions it is very difficult to keep the overview.  Thus I would suggest simplifying the labeling by indicating the activation or inhibition of pathways (for instance ERK+/-) instead. Color coding might be another option to ease the reader through the complex figures, in particular Fig. 4, which is very hard to follow and reconcile.

             We followed the advice of the reviewer and referred to the pharmacological reagents as either activators or inhibitors, and for enhanced visual clarity in figures these activators and inhibitors are labelled green and red, respectively.

             In Fig. 4 many comparisons are made, but I wonder if all of these are relevant for the main theme of the work, namely the elucidation of CD40L reverse signalling in restricting dendrite growth in MSNs. While the pharmacological effects on the phopho-status of the respective kinases with Fc only are important controls, the reported changes do not reflect CD40L reverse signalling effects and make it hard to keep an overview. I would suggest to focus on comparisons in the presence of Fc-CD40L, as this is relevant for dendrite restriction of MSNs and better reflects the potential reciprocal effects of JNK/PKC/ERK depending on their activation status. Fc controls could be shown as a supplement instead. This would also allow for easier comparison with the respective dendrite phenotypes shown in the supplement.

             We agree with the reviewer that, even though the comparisons with Fc are important, they distract from the main focus of the work and make the results section longer and more complex than it needs to be. We have removed the Fc results from main figures and simplified the text accordingly. The Fc results are now included in a supplemental figure.   

             Detailed statistics (including n numbers) are missing and should be added at least as a supplemental file. Otherwise it is not clear if the data without shown is significantly changed or not. For example in Fig. 2, does Anisomycin significantly reduce dendrite length with Fc-CD40 compared to control? Moreover, the control plots for Fc and CD40-Fc in Fig. 2.c,e,g show the same data points. This should be at least mentioned in the legend. Exact n numbers should be stated in the legend as well.

            We included the n numbers in the supplemental material.  

            Statistics not shown in figure legends are stated in the supplemental material. 

            The legend for Suppl. Fig. 1 is missing and statistics should be included there.

             Included.

            In line 519 the authors state that “the function of JNK is little modulated by the activation state of PKC or ERK1/ERK2.” and conclude that activation of JNK is decisive for the described activity and dendrite growth restraint. However, PKC manipulation equally affects the pJNK status, and dendrite growth restriction with PMA is not strongly affected by JNK/ERK manipulation (no significances are given in Suppl. Fig,1). While I do not argue with the overall conclusion of the authors, clarification of this point would be helpful.

             We agree with the reviewer that both JNK and PKC play decisive roles in controlling neurite growth. We have made clear this important point in the revised text.

 Minor  points:
1. In the initial part of the introduction references should be cited to support the authors’ statements (e.g. lines 39-44).

             We added the reference to support our statement.

  1. Typo in line 70 “CD40”-L 

            Done

  1. Fig. 2 does not show >60 dots. Please check and update statistics.

             In the corrected version the number of neurons counted (n) for all conditions are indicated. All statistics were made with the exact number of cells counted.

  1. In lines 277-279 the authors claim that basal ERK activity favors growth, while its activation via Fis shows a reduction. Thus ERK pathway activity seems to somehow reduce growth in both cd40-/- conditions. The authors do not explain this effect. Is this due to PKC phosphorylation?

             This is a very interesting question. We do not have a conclusive explanation, but as the reviewer suggests the most plausible explanation is related to the fact that activation of ERK with Fis modulates the phosphorylation levels of PKC and JNK. When CD40L is not activated by CD40-Fc, as showed in Supp Fig 2, activation of ERK increased pJNK and pPKC (white bars, in presence of Fc), and the activation of these proteins restricts neurite growth.

  1. line 348:“The  inhibition of JNK that suppresses the neurite growth is only restored by activation of PKC” sounds as if JNK inhibition suppresses dendrite growth, which is not the case           

            The reviewer is right. The correct sentence now in the text is "The inhibition of JNK that suppresses the control over neurite growth is only restored by activation of PKC".

  1. Line 426 typo “effects” should be singular

             Done

  1. Line 568 Typo “MNSs”

             Corrected

  1. Fig 5: To make it more clear, the strength of phosphorylation might be indicated by the stroke width of the arrows. A color code might also help to follow the figure

            To facilitate understanding of the figures, we have labelled activators green and inhibitors red. We have coloured arrows similarly to indicate increases or reduces the phosphorylation. Because the phosphorylation levels obtained by different antibodies are not comparable, we did not indicate the strength of phosphorylation by the width of the arrows.

Reviewer 3 Report

This study demonstrates that the activation of CD40L-mediated revers -signalling in cultured mice neurones increases the phosphorylation of PKC, ERK1/2 and JNK cascade and that the pharmacological perturbation of these pathways either individually or in combination, affect neurite outgrowth – albeit in a cell specific manner. An extensive battery of pharmacological and immunoprecipitation assays was used to demonstrate both the dependence and interdependence of these pathways on the growth of cultured, vertebrate excitatory and inhibitory neurones in growing cultures.

This study contains an impressive amount of data which could be potentially significant and of importance to this journal. Unfortunately, through this paper is not well written; the writing is dense, littered with jargon. The rationale and the logical of inquiry are not well spelled out. This shortcoming thus seriously impacts the potential outcome of this study and makes the conclusions drawn difficult to deduce. The authors need to stream line this paper and explicitly state what the rationale of this study was and the problems that they had wished to investigate.

Although the figures are well presented and the data look convincing, but neither does the text nor the legends do justice to their completed work. It is therefore difficult to decipher what the questions were, and how does the data present will do justice to the completed work. No rationale or explanation is provided as to why any particular pathway was chosen and why certain pharmacological tools were used. Moreover, the effects of various pharmacological agents on any given signalling pathway, their off-target effects and impact on other signalling pathways/functions should have been considered and discussed. This study is not well designed nor is it presented in a logical manner. For instance, it is unclear if any of these compounds would also affect axonal growth, cellular viability, synaptic connectivity and function? Do we know the extent to which the growth and the dendritic branching patterns could be altered by any of these signaling pathways? Impact of these agent on glia cells and their contributions to the extent of neuronal growth etc. are other important questions that would need to be addressed discussed.

The experiments and the experimental design would need to be setup better; as it stands, most of these are just thrown in and lack a clear rationale. These shortcomings make it very difficult to warrant a full comprehensive review of this paper.

Author Response

This study demonstrates that the activation of CD40L-mediated revers -signalling in cultured mice neurones increases the phosphorylation of PKC, ERK1/2 and JNK cascade and that the pharmacological perturbation of these pathways either individually or in combination, affect neurite outgrowth – albeit in a cell specific manner. An extensive battery of pharmacological and immunoprecipitation assays was used to demonstrate both the dependence and interdependence of these pathways on the growth of cultured, vertebrate excitatory and inhibitory neurones in growing cultures.

This study contains an impressive amount of data which could be potentially significant and of importance to this journal. Unfortunately, through this paper is not well written; the writing is dense, littered with jargon. The rationale and the logical of inquiry are not well spelled out. This shortcoming thus seriously impacts the potential outcome of this study and makes the conclusions drawn difficult to deduce. The authors need to stream line this paper and explicitly state what the rationale of this study was and the problems that they had wished to investigate.

             We edited the Introduction indicating more explicitly the aim of the manuscript. We also included a brief description of significance of the results obtained using the different experimental approaches. Moreover, to make the manuscript more accessible we edited, shortened and focused the Results section and used colour labelling to facilitate visual comprehension of the data.  

 Although the figures are well presented and the data look convincing, but neither does the text nor the legends do justice to their completed work. It is therefore difficult to decipher what the questions were, and how does the data present will do justice to the completed work. No rationale or explanation is provided as to why any particular pathway was chosen and why certain pharmacological tools were used. Moreover, the effects of various pharmacological agents on any given signalling pathway, their off-target effects and impact on other signalling pathways/functions should have been considered and discussed.

             We included a sentence in the Introduction justifying the reasons for analyzing these signalling pathways. Basically, this is based on the fact that in excitatory neurons the downstream signalling engaged after activation of CD40L reverse signalling involves these signalling pathways.

            As stated at the beginning of section 3.2, the pharmacological reagents used are specific activators and inhibitors of respective signalling pathways with the concentrations specified in their datasheets. We agree with the reviewer we cannot ignore the fact that these pharmacological drugs may have off-target effects. This has been added to the text.

This study is not well designed nor is it presented in a logical manner. For instance, it is unclear if any of these compounds would also affect axonal growth, cellular viability, synaptic connectivity and function? 

            In previously published work on pyramidal neurons, we showed that CD40L reverse signalling regulates the growth of both dendrites and axons. We also showed that in MSNs the regulation mediated by CD40L reverse signalling affects all neural projections. In cultured MSNs axons cannot be easily distinguished from dendrites. For this reason, in the revised text we used the term neurite, because this refers to any projection from the soma (axon or dendrite).

            We did not observe any differences in cell density or in morphological characteristics suggestive of altered neuron viability with any of the treatments used. This observation is mentioned in the revised text.

            Because the aim of this study was to determine the downstream signalling events responsible for the effects of CD40L reverse signalling on neurite growth, we performed experiments to address this question. After determining that CD40L reverse signalling activates JNK, PKC and ERK1/ERK2 we manipulated these pathways with activators and inhibitors to determine their effects on neurite growth. Although the effects of these reagents on synaptic connectivity and neuronal function could be interesting, these effects were beyond the scope of the current study.  

Do we know the extent to which the growth and the dendritic branching patterns could be altered by any of these signaling pathways?

            As mentioned in the Introduction (references 3 to 12), these signalling pathways have been shown to regulate both the length and branching neurites. While we focused on neurite length to simplify the current study, we nonetheless obtained branching data for all experimental treatments. These data showed that the effects of different experimental treatments on neurite length and on branching were highly correlated. Similar statistical differences were observed for length and branching among the different experimental treatments. These data are plotted in a new supplementary figure together with the statistical differences.  

Impact of these agent on glia cells and their contributions to the extent of neuronal growth etc. are other important questions that would need to be addressed discussed. The experiments and the experimental design would need to be setup better; as it stands, most of these are just thrown in and lack a clear rationale. These shortcomings make it very difficult to warrant a full comprehensive review of this paper.

             As stated, the aim of this current study was to ascertain the downstream signalling responsible for the regulation of neurites growth from MSNs after activation of CD40L reverse signalling. Our work is based on a previous study in which we concluded that CD40L reverse signalling is a major regulator of neurite growth from MSNs. In that manuscript we discussed the probable function of glial cells as a possible source of CD40 for activating CD40L reverse signalling. While we do not feel it is justified to repeat this speculation here, we have referred the reader to this earlier study.

            We have clarified the rationale for the experimental design in each section. We hope these comments will make our manuscript more accessible to the reader.

Round 2

Reviewer 1 Report

The authors addressed my comments

Reviewer 2 Report

I think the clarity of the results is vastly improved and the manuscript can be accepted in the present form. Some minor text editing for English grammar would be good though.